# Harnessing Patient Connectivity for Medical Classification under Missing Values

## Abstract

Various machine learning techniques have been developed to classify patients for disease diagnosis using medical tabular data. Due to the presence of missing values in the medical tabular data, these techniques commonly impute the missing values before applying classifiers. However, most existing techniques classify patients solely based on each patient's individual features, overlooking the potential benefits of using similarities among patients to improve both imputation and classification. To address this limitation, we introduce Graph-based Feature-Attentive Classifier under Missingness (G-FACM), a novel framework for classification on medical tabular data. G-FACM constructs feature-attentive k-nearest neighbor (kNN) graphs to seamlessly integrate graph data imputation methods with medical tabular classification. The key idea is to construct a kNN graph among patients by prioritizing features that are most important for classification. Our extensive experimental results demonstrate that G-FACM successfully bridges the gap between graph data imputation methods and medical tabular classification, achieving state-of-the-art performance across various medical tabular datasets.

## 1 Introduction

Recent progress in machine learning technology has led to substantial strides in the medical domain (Kononenko, 2001; Giger, 2018; Shehab et al., 2022; Joshi et al., 2024). Among various types of data in the medical domain, tabular data is one of the most widely used forms, comprising numerical and categorical features for each patient. Many researchers have utilized machine learning frameworks on medical tabular data to classify patients for disease diagnosis (Rahman & Davis, 2013; Liu et al., 2023). The main challenge in handling medical tabular data is that it often contains missing values due to various factors, such as privacy concerns or incomplete data collection. In this paper, we tackle the classification of patients on medical tabular data with missing values.

To classify patients in medical tabular data containing missing values, imputation techniques are necessary to fill in the missing values. This is because most classifiers assume fully observed input data. Traditionally, simple imputation techniques, such as zero and mean imputation, have been widely used for medical tabular data (Graham et al., 1997; Schafer & Graham, 2002). Recently, deep learning-based imputation techniques (Mattei & Frellsen, 2019; You et al., 2020; Zhong et al., 2023) have demonstrated powerful performance on tabular data, making them an effective approach for medical tabular data. After filling in missing values through the imputation methods, a Multi-Layer Perceptron (MLP) is commonly employed to classify each patient based on the complete data (Sivasankari et al., 2022; Levin et al., 2022).

Medical tabular data typically contains two types of features (Remeseiro & Bolon-Canedo, 2019): (1) class-discriminative features that differentiate among classes and (2) non-discriminative features that have the same distribution regardless of class. For instance, in Alzheimer's disease data (Petersen et al., 2010), the score of a logical memory test or a standard clinical rating can serve as a key discriminative feature for identifying the disease, while many non-discriminative features, such as the years from the first measurement and the site where data was collected, also exist. Specifically, patients with a particular disease tend to exhibit similar class-discriminative features. Therefore, the classification of a patient can be aided by considering patients who have class-discriminative features similar to those of the patient in question.

Meanwhile, imputation methods developed for graph-structured data (Taguchi et al., 2021; Rossi et al., 2022; Um et al., 2023) have garnered significant attention due to their remarkable effectiveness in classification, even in the presence of high rates of missing values. These graph data imputation methods incorporate graph neural networks (GNNs) within their frameworks to perform classification tasks. In many real-world graph-structured datasets, there is homophily, which refers to the tendency for nodes to be connected when they belong to the same class or have similar feature values. Based on homophily (McPherson et al., 2001), graph data imputation methods leverage the valuable information in each node's neighbors, leading to outstanding performance in downstream tasks. Although tabular medical data does not have predefined connectivity among patients, connecting patients with similar class-discriminative features can promote homophily, where connected patients are more likely to belong to the same class.

To this end, we propose a novel framework called Graph-based Feature-Attentive Classifier under Missingness (G-FACM) for classification on medical tabular data. G-FACM constructs a kNN graph on medical tabular data with a focus on class-discriminative features. G-FACM first trains a feature-wise attention network to infer the influence of each feature value on classification. Using the trained network, it then computes the importance of each feature across all patients. Based on the resulting feature importance, G-FACM constructs a kNN graph tailored to the classification task. Finally, by introducing the constructed graph into graph data imputation methods, we enable effective classification on medical tabular data. Despite its simplicity, this feature-attentive graph construction significantly improves classification performance over existing kNN graph construction methods, demonstrating the seamless integration of graph data imputation and medical tabular classification.

The main contributions of our work are summarized as: (1) To the best of our knowledge, this work is the first attempt to apply graph data imputation methods to tabular data. (2) Based on the nature of medical tabular data, our G-FACM builds a kNN graph that is attentive to class-discriminative features, bridging graph data imputation methods and medical tabular data. (3) We demonstrate that our G-FACM using feature-attentive kNN graphs significantly outperform existing state-of-the-art methods in medical tabular classification and G-FACM can also provide valuable medical insights.

## 2 RELATED WORK

### 2.1 TABULAR DATA IMPUTATION

Since missing data is a pervasive problem across various domains, handling missing data has long been a prominent area of research in machine learning (Allison, 2009; Lin & Tsai, 2020). For missing data imputation on tabular data, simple imputation methods such as zero imputation (Schafer & Graham, 2002), mean imputation (Graham et al., 1997), and kNN imputation (Troyanskaya et al., 2001), as well as statistical methods (Van Buuren & Groothuis-Oudshoorn, 2011), have been widely used. With the advancement of deep learning models, deep learning-based approaches have gained popularity due to their effectiveness for accurate imputation. GAIN (Yoon et al., 2018) adopts a Generative Adversarial Nets (GAN) (Goodfellow et al., 2014) framework to generate missing values in tabular datasets. MIWAE (Mattei & Frellsen, 2019) is a framework that enhances Importance-Weighted AutoEncoder (IWAE) (Burda et al., 2015) by introducing a lower bound on the likelihood of observed data to the original objective of IWAE. Recently, graph-based imputation methods, including GRAPE (You et al., 2020) and IGRM (Zhong et al., 2023), have been proposed. These graph-based methods transform a given tabular dataset into a bipartite graph, where nodes consist of sample nodes and feature nodes. By predicting the edge weight between a sample node and a feature node on this bipartite graph, the graph-based methods estimate missing values in the tabular dataset. To perform classification tasks after imputation processes, sample-level classifiers, such as an MLP classifier, are commonly applied to the data completed by various imputation techniques. However, sample-level classifiers cannot leverage the relationships among samples, which can play a crucial role in classification tasks.

### 2.2 GRAPH DATA IMPUTATION

Several methods tackle the reconstruction of missing values in graph-structured data by minimizing the reconstruction error between the observed values and their reconstructed values (Monti et al., 2017; Chen et al., 2020). However, since node classification is a primary task in graph learning (Xiao et al., 2022), many approaches have been developed to address node classification with missing values rather than focusing on the accurate reconstruction of missing values. These approaches can

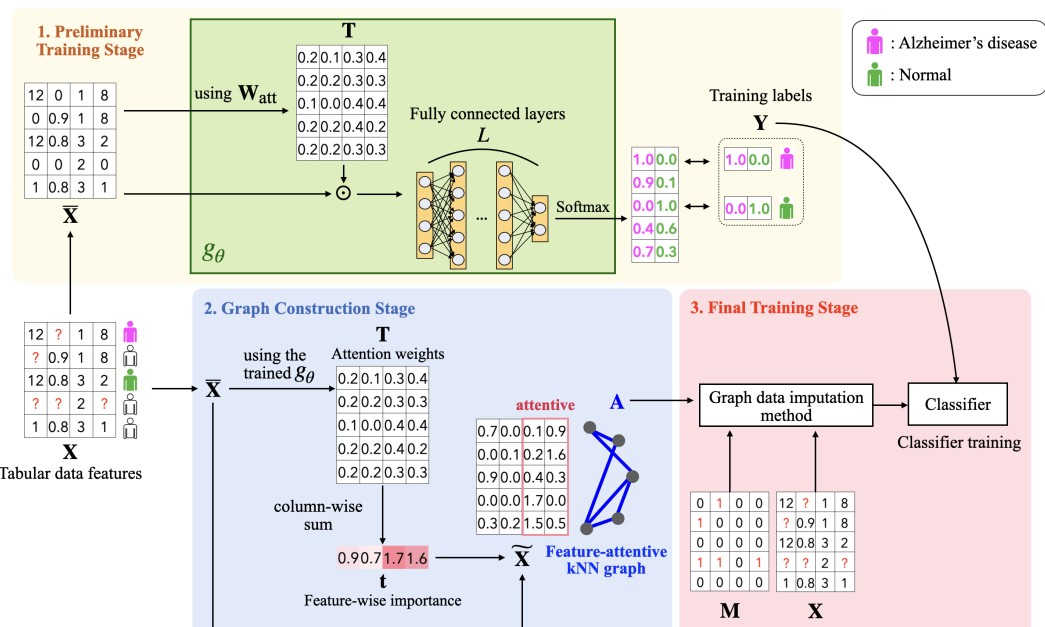

Figure 1: A brief overview of G-FACM: In the preliminary training stage, $g_\theta$, an MLP classifier with an attention mechanism, is first trained using supervised learning. In the graph construction stage, we utilize the trained $g_\theta$ to compute $\mathbf{t}$, which represents feature-wise importance. Using $\mathbf{t}$, this stage constructs a kNN graph that focuses on class-discriminative features. Finally, $\mathbf{A}$, the adjacency matrix of the kNN graph, is provided to the final training stage, enabling graph data imputation methods to be trained and to perform a classification task.

be categorized into propagation-based methods and GNN architecture-based methods. Propagation-based methods, including FP (Rossi et al., 2022) and PCFI (Um et al., 2023), impute missing values through the iterative propagation of observed values on a graph. While preserving the observed values, these methods update missing values by repeatedly aggregating values from neighboring nodes. After imputation, propagation-based methods, employ GNN to perform node classification tasks. As for GNN architecture-based methods, such as GCNMF (Taguchi et al., 2021) and PaGNN (Jiang & Zhang, 2020), these approaches introduce new GNN architectures to learn from graph-structured data with partially observed feature values. These methods can be regarded as integrating imputation and classification within a unified GNN architecture. Thus, in this paper, we collectively refer to both propagation-based and GNN architecture-based approaches as graph data imputation methods. While all graph data imputation methods require predefined connectivity among samples, our G-FACM enables these algorithms to operate on tabular data classification by constructing a new graph primarily based on class-discriminative features.

## 3 PROPOSED METHOD

### 3.1 PROBLEM SETUP

We consider a medical tabular dataset containing missing feature values. We let $\mathbf{X}_{og} \in \mathbb{R}^{N \times F_{og}}$ be the feature matrix of the given medical tabular dataset, where $N$ and $F_{og}$ denote the number of samples (patients) and the number of features, respectively. $\mathbf{M}_{og} \in \{0,1\}^{N \times F_{og}}$ denotes a binary mask where values of 1 indicate the location of missing values. These features consist of numerical and categorical features. To employ imputation techniques, we convert each categorical feature into its dummy variables. This process yields $\mathbf{X} \in \mathbb{R}^{N \times F}$ and $\mathbf{M} \in \{0,1\}^{N \times F}$ from $\mathbf{X}_{og}$ and $\mathbf{M}_{og}$, respectively, where $F$ represents the sum of the number of given numerical features and the number of dummy variables. Let $\mathbf{Y} = [y_1, \ldots, y_N]^\top$ be the labels of samples and $y_i \in \{1, \ldots, C\}$, where $y_i$ denotes the disease-related class label of the $i$-th sample and $C$ denotes the number of classes. We assume that labels are given for only a subset of the samples (*i.e.*, the training samples). The remaining samples, which are not used for training, are unlabeled (*i.e.*, the validation and test samples). The goal of medical classification is to predict the classes of the test samples based on $\mathbf{X}$, which contains missing values and the partially available labels for the training samples.

## 3.2 OVERVIEW OF G-FACM

We propose a novel approach called Graph-based Feature-Attentive Classifier under Missingness (G-FACM), designed to adapt graph data imputation techniques for medical tabular data containing missing values. While graph data imputation methods require predefined connectivity among data points, medical tabular data typically lacks inherent connectivity. To transfer the powerful performance of graph data imputation to the medical tabular domain, G-FACM constructs the connectivity among patients. G-FACM is designed to construct this graph structure mainly based on class-discriminative features to assist graph data imputation methods in performing classification.

Figure 1 provides a brief overview of G-FACM. G-FACM consists of three stages: a preliminary training stage, a graph construction stage, and a final training stage. In the preliminary training stage, $g_\theta$, an MLP classifier with an attention mechanism, is trained using supervised learning. After training, all samples are passed through the trained $g_\theta$ to obtain the feature-wise attention weights for each sample. These attention weights are then summed across the samples in a feature-wise manner, producing feature-wise attention weights. In the graph construction stage, a kNN graph is built using weighted cosine similarity based on the feature-wise attention weights, making the kNN graph attentive to class-discriminative features. In the final training stage, using this kNN graph, graph data imputation models are trained utilizing GNN-based frameworks to classify samples.

## 3.3 PRELIMINARY TRAINING STAGE

Given $\mathbf{X} \in \mathbb{R}^{N \times F}$, the feature matrix of a medical tabular dataset, we first produce $\overline{\mathbf{X}} \in \mathbb{R}^{N \times F}$ from $\mathbf{X}$ by imputing missing values with zeros. Using $\overline{\mathbf{X}}$, the preliminary training stage then trains $g_\theta$, an MLP classifier with an attention mechanism. Specifically, we compute attention weights $\mathbf{T} \in \mathbb{R}^{N \times F}$ as follows:

$$\mathbf{T}_{i,j} = \frac{\exp\left(((\overline{\mathbf{X}}_{i,:})\mathbf{W}_{\text{att}})_j\right)}{\sum_{k=1}^{F} \exp\left(((\overline{\mathbf{X}}_{i,:})\mathbf{W}_{\text{att}})_k\right)}, \tag{1}$$

where $\mathbf{W}_{\text{att}} \in \mathbb{R}^{F \times F}$ is a trainable weight matrix and $\overline{\mathbf{X}}_{i,:}$ denotes the $i$-th row of $\overline{\mathbf{X}}$. Here, $\mathbf{T}_{i,j}$ represents the attention weight for the $j$-th feature of the $i$-th sample, and the softmax function ensures that the sum of attention weights across all features for each sample equals 1 (*i.e.*, $\sum_{j=1}^{F} \mathbf{T}_{i,j} = 1$). We then apply the attention weights $\mathbf{T}$ to $\overline{\mathbf{X}}$ as follows:

$$\overline{\mathbf{X}}^{\text{att}} = \overline{\mathbf{X}} \odot \mathbf{T}, \tag{2}$$

where $\odot$ denotes element-wise multiplication.

$g_\theta$, consisting of $L$ layers, then processes the attention-weighted feature matrix $\overline{\mathbf{X}}^{\text{att}}$ through a series of fully connected layers in a sample-wise manner, applying linear transformations followed by non-linear activations such as ReLU. Formally,

$$\mathbf{H}^{(l)} = \sigma(\mathbf{H}^{(l-1)}\mathbf{W}^{(l)} + \mathbf{b}^{(l)}), \quad l = 1, \dots, L-1 \tag{3}$$

where $\mathbf{H}^{(l)}$ represents the output of the $l$-th layer, $\mathbf{W}^{(l)} \in \mathbb{R}^{d_{l-1} \times d_l}$ and $\mathbf{b}^{(l)} \in \mathbb{R}^{d_l}$ are the weight matrix and bias vector of the $l$-th layer, respectively, and $\sigma(\cdot)$ denotes the activation function (*e.g.*, ReLU). Here, $\mathbf{H}^{(0)} = \overline{\mathbf{X}}^{\text{att}}$, the input to the first layer, with $d_0 = F$.

This process continues through all the hidden layers until the final layer, where the output logits $\hat{\mathbf{Y}} \in \mathbb{R}^{N \times C}$ are computed as:

$$\hat{\mathbf{Y}} = \mathbf{H}^{(L-1)}\mathbf{W}^{(L)} + \mathbf{b}^{(L)}, \tag{4}$$

where $\mathbf{W}^{(L)} \in \mathbb{R}^{d_{L-1} \times C}$ and $\mathbf{b}^{(L)} \in \mathbb{R}^C$ are the weight matrix and bias vector of the output layer, respectively. Finally, a softmax function is applied to the logits for each sample to produce the predicted class probabilities. $g_\theta$ is trained using cross-entropy loss, computed between the one-hot encoded labels from the training sample labels $\mathbf{Y}$ and the predicted probabilities from $\hat{\mathbf{Y}}$.

## 3.4 GRAPH CONSTRUCTION STAGE

After $g_\theta$ is trained through the training stage, $\mathbf{T}$ can provide the importance of each feature for each sample. Thus, we feed $\overline{\mathbf{X}}$ into the trained $g_\theta$ and obtain the attention weights $\mathbf{T}$. Since G-FACM requires feature-wise importance representing the degree to which each feature contributes to classifying the classes, we calculate the feature-wise importance $\mathbf{t} \in \mathbb{R}^F$ by summing $\mathbf{T}$ across

the samples as $\mathbf{t}_j = \sum_{i=1}^{N} \mathbf{T}_{i,j}$. This feature-wise importance $\mathbf{t}$ reflects the degree to which each feature is class-discriminative.

To construct a kNN graph using $\mathbf{t}$, we first normalize each sample in $\overline{\mathbf{X}}$ and weight the features according to $\mathbf{t}$ as follows:

$$\widetilde{\mathbf{X}}_{i,j} = (\mathbf{t}_j)^\alpha \cdot \frac{\overline{\mathbf{X}}_{i,j}}{\|\overline{\mathbf{X}}_{i,:}\|_2}, \quad \text{for } i = 1, \ldots, N, \tag{5}$$

where $\widetilde{\mathbf{X}} \in \mathbb{R}^{N \times F}$, $\alpha > 0$ is a hyperparameter that controls the influence of feature importance, and $\|\overline{\mathbf{X}}_{i,:}\|_2$ is the L2 norm of the $i$-th row of $\overline{\mathbf{X}}$.

Given an arbitrary matrix $\mathbf{B} \in \mathbb{R}^{a \times b}$, we define $\text{kNN}(\cdot) : \mathbb{R}^{a \times b} \rightarrow \{0, 1\}^{a \times a}$ as a function that generates an adjacency matrix of the row-wise kNN graph (*i.e.*, the kNN graph among rows) based on cosine similarity. We build the kNN graph among samples by

$$\mathbf{A} = \text{kNN}(\widetilde{\mathbf{X}}), \tag{6}$$

where $\mathbf{A} \in \{0, 1\}^{N \times N}$ denotes the connections among samples, with values of 1 indicating connected samples. Since $\widetilde{\mathbf{X}}$ is calculated using $\mathbf{t}$, $\mathbf{A}$ can be constructed with a primary focus on class-discriminative features rather than non-discriminative ones.

### 3.5 FINAL TRAINING STAGE

Although the given medical dataset does not have any predefined connectivity, G-FACM can provide $\mathbf{A}$ to graph data imputation methods that require the connectivity among samples as well as a feature matrix $\mathbf{X}$ and a mask $\mathbf{M}$ indicating the location of missing values. Thus, $\mathbf{A}$, the output of the graph construction stage, enables the use of graph data imputation methods on medical tabular data. Graph data imputation methods incorporate GNN frameworks. Specifically, while GCNMF (Taguchi et al., 2021) and PaGNN (Jiang & Zhang, 2020) are GNN architectures, propagation-based methods, including FP (Rossi et al., 2022) and PCFI (Um et al., 2023), use downstream GNNs to perform classification tasks. For patient classification, GNN models in graph data imputation methods are trained using $\mathbf{A}$ to classify samples, thereby transferring their powerful performance from the graph domain to the medical tabular domain.

## 4 EXPERIMENTS

### 4.1 DATASETS

We conduct experiments on six medical tabular datasets, all of which initially contain missing data, as follows: Echocardiogram (Asuncion et al., 2007), Duke Breast Cancer (Saha et al., 2018), ABIDE (Di Martino et al., 2014), ADNI QT-PAD (Petersen et al., 2010), ADNI TADPOLE (Petersen et al., 2010), and Diabetes 130-US (Asuncion et al., 2007). The datasets have missing data rates of 2.59%, 11.94%, 52.52%, 22.29%, 27.31%, and 4.03%, respectively. In addition, we use Diabetes CDC (Asuncion et al., 2007), a large-scale dataset with 253,680 samples. Detailed information on these datasets is provided in Appendix A.1.

### 4.2 COMPARED METHODS

We compare G-FACM with seven tabular data imputation methods on medical tabular datasets. These methods are categorized into two groups: (1) conventional methods: zero imputation (Schafer & Graham, 2002), mean imputation using the feature-wise mean (Graham et al., 1997), and kNN imputation (Troyanskaya et al., 2001); and (2) state-of-the-art deep learning-based methods: GAIN (Yoon et al., 2018), MIWAE (Mattei & Frellsen, 2019), GRAPE (You et al., 2020), and IGRM (Zhong et al., 2023). For graph data imputation methods, we employ GCNMF (Taguchi et al., 2021), PaGNN (Jiang & Zhang, 2020), FP (Rossi et al., 2022), and PCFI (Um et al., 2023). As the default setting for G-FACM, FP is utilized as a graph data imputation method. That is, unless otherwise specified, we use FP as the graph data imputation method for G-FACM. We provide URL links for all compared methods in Appendix C.

### 4.3 EXPERIMENTAL SETUP

To evaluate the performance of imputation methods in medical classification with missing data, we compare classification performance on medical tabular data containing missing values. We generate

Table 1: Classification results measured by Micro-F1 score (%). Standard deviation errors are given. OOM denotes an out-of-memory error.

| Method | Echocardiogram | Duke Breast Cancer | ABIDE | ADNI QT-PAD | ADNI TADPOLE | Diabetes 130-US |
|---|---|---|---|---|---|---|
| Zero | $75.33_{\pm 3.06}$ | $74.80_{\pm 3.91}$ | $91.30_{\pm 0.54}$ | $78.22_{\pm 0.99}$ | $77.94_{\pm 1.24}$ | $53.66_{\pm 0.77}$ |
| Mean | $73.00_{\pm 4.88}$ | $72.76_{\pm 7.02}$ | $68.43_{\pm 2.13}$ | $78.35_{\pm 1.53}$ | $76.89_{\pm 1.49}$ | $53.76_{\pm 0.33}$ |
| kNN | $77.00_{\pm 3.71}$ | $76.53_{\pm 0.82}$ | $90.45_{\pm 1.17}$ | $80.39_{\pm 1.30}$ | $76.68_{\pm 1.07}$ | $53.92_{\pm 0.86}$ |
| GAIN | $68.67_{\pm 4.99}$ | $76.31_{\pm 1.32}$ | $89.30_{\pm 1.81}$ | $77.46_{\pm 1.22}$ | $75.46_{\pm 1.22}$ | $53.58_{\pm 0.59}$ |
| MIWAE | $69.43_{\pm 6.25}$ | OOM | $64.33_{\pm 0.93}$ | OOM | OOM | OOM |
| GRAPE | $75.00_{\pm 0.81}$ | $75.90_{\pm 1.35}$ | $91.61_{\pm 0.89}$ | $79.72_{\pm 1.80}$ | $77.46_{\pm 1.81}$ | $53.70_{\pm 0.95}$ |
| IGRM | $69.33_{\pm 8.21}$ | $75.13_{\pm 1.58}$ | $66.38_{\pm 1.85}$ | $78.60_{\pm 1.44}$ | $76.80_{\pm 1.59}$ | $53.49_{\pm 0.74}$ |
| G-FACM (ours) | $\mathbf{89.00_{\pm 2.71}}$ | $\mathbf{76.62_{\pm 0.67}}$ | $\mathbf{91.69_{\pm 1.14}}$ | $\mathbf{83.75_{\pm 0.99}}$ | $\mathbf{80.01_{\pm 1.22}}$ | $\mathbf{54.65_{\pm 1.11}}$ |

Table 2: Comparison of feature-attentive kNN graph construction and typical graph construction algorithms in terms of Micro-F1 score for medical classification. FC, SIM_FC, kNN, and ATT_kNN represent an unweighted fully connected graph, a fully connected graph with feature similarity weights, a typical kNN graph, and our feature-attentive kNN graph. G-FACM models with different graph construction algorithms are evaluated. Improvement (%) denotes the improvement percentage, representing the percentage improvement of ATT_kNN over kNN.

| Dataset | Echocardiogram | Duke Breast Cancer | ABIDE | ADNI QT-PAD | ADNI TADPOLE | Diabetes 130-US |
|---|---|---|---|---|---|---|
| G-FACM with FC | $67.67_{\pm 1.33}$ | $77.08_{\pm 0.70}$ | $49.62_{\pm 1.80}$ | $32.34_{\pm 0.14}$ | $28.96_{\pm 0.57}$ | OOM |
| G-FACM with SIM_FC | $67.67_{\pm 1.33}$ | $77.08_{\pm 0.70}$ | $49.62_{\pm 1.80}$ | $32.34_{\pm 0.14}$ | $28.96_{\pm 0.57}$ | OOM |
| G-FACM with kNN | $85.67_{\pm 4.67}$ | $75.38_{\pm 2.82}$ | $90.65_{\pm 1.51}$ | $83.31_{\pm 1.25}$ | $78.90_{\pm 0.94}$ | $53.03_{\pm 0.85}$ |
| G-FACM with ATT_kNN (ours) | $\mathbf{89.00_{\pm 2.71}}$ | $\mathbf{76.62_{\pm 0.67}}$ | $\mathbf{91.69_{\pm 1.14}}$ | $\mathbf{83.75_{\pm 0.99}}$ | $\mathbf{80.01_{\pm 1.22}}$ | $\mathbf{54.65_{\pm 1.11}}$ |
| Improvement | +3.89% | +1.64% | +1.14% | +0.53% | +1.41% | +3.05% |

five random splits for training, validation, and test samples with proportions of 0.1, 0.1, and 0.8, respectively. To evaluate classification performance, we measure the average Micro-F1 score across the five splits. For the six tabular data imputation methods except for GRAPE, we employ MLP classifiers on the imputed feature matrices to perform classification. Since GRAPE has an integrated version that includes a classifier, we use that version for classification. Graph data imputation methods are categorized into two approaches: (1) single-stage, including GCNMF and PaGNN, and (2) two-stage, including FP and PCFI. While the single-stage methods perform imputation and classification within a single framework, we utilize GCNs (Kipf & Welling, 2016) as downstream GNNs for the two-stage methods. For G-FACM, we employ grid search to tune $\alpha$ in Eq. (5) and $k$ in the kNN graph construction in Eq. (6). $k$ and $\alpha$ are searched within $\{1, 3, 5, 10\}$ and $\{0.25, 0.5, 0.75, 1\}$, respectively, using the validation sets. We provide further details on experiments in Appendix A.2.

### 4.4 COMPARISON WITH STATE-OF-THE-ART METHODS

On medical tabular datasets containing initially missing values, we compare the classification performance of G-FACM against tabular data imputation methods. Table 1 demonstrates the classification performance comparison among the methods, measured by Micro-F1 score (%). As shown in the table, G-FACM achieves state-of-the-art performance across all datasets. Moreover, the performance gains of our best method over the previous state-of-the-art methods are significant. For example, on Echocardiogram, Alzheimer's Disease Neuroimaging Initiative (ADNI) QT-PAD, and ADNI TAD-POLE, the gains are 15.58%, 4.18%, and 2.66%, respectively. Furthermore, we observe that deep learning-based tabular imputation methods, except for GAIN, suffer from out-of-memory errors, indicating poor scalability. In contrast, our method does not suffer from out-of-memory errors, demonstrating the memory efficiency of G-FACM.

### 4.5 COMPARISON OF FEATURE-ATTENTIVE kNN GRAPH CONSTRUCTION AND EXISTING GRAPH CONSTRUCTION ALGORITHMS

To investigate the source of G-FACM's outstanding performance, we conduct experiments comparing G-FACM with feature-attentive kNN graph construction to G-FACM with typical graph construction algorithms, including an unweighted fully connected graph (denoted as FC), a fully connected graph with feature similarity weights (denoted as SIM_FC), and a typical kNN graph. Table 2 presents the comparison results. As shown in the table, feature-attentive kNN graph construction (denoted as ATT_kNN) significantly improves the performance of G-FACM compared to its use with typical graph construction algorithms. Notably, ATT_kNN consistently outperforms typical graph construction algorithms across all datasets. This indicates that feature-attentive kNN

Table 3: Classification performance for varying label rates, measured by Micro-F1 score (%). OOM denotes an out-of-memory error.

| Dataset | Echocardiogram | | | Duke Breast Cancer | | | ABIDE | | |
|---|---|---|---|---|---|---|---|---|---|
| Label rate | 5% | 10% | 20% | 5% | 10% | 20% | 5% | 10% | 20% |
| Zero | $67.50_{\pm10.15}$ | $75.33_{\pm3.06}$ | $78.11_{\pm2.82}$ | $75.57_{\pm1.29}$ | $74.80_{\pm3.91}$ | $77.48_{\pm0.80}$ | $88.67_{\pm0.74}$ | $91.30_{\pm0.54}$ | $91.17_{\pm0.75}$ |
| Mean | $65.00_{\pm4.90}$ | $73.00_{\pm4.88}$ | $73.21_{\pm6.13}$ | $75.39_{\pm2.38}$ | $72.76_{\pm7.02}$ | $76.82_{\pm1.40}$ | $62.96_{\pm4.02}$ | $68.43_{\pm2.13}$ | $71.63_{\pm1.62}$ |
| kNN | $71.56_{\pm13.67}$ | $77.00_{\pm3.71}$ | $76.98_{\pm8.72}$ | $75.54_{\pm2.02}$ | $76.53_{\pm0.82}$ | $77.42_{\pm0.68}$ | $88.25_{\pm0.75}$ | $90.45_{\pm1.17}$ | $91.04_{\pm0.65}$ |
| GAIN | $66.25_{\pm9.09}$ | $68.67_{\pm4.99}$ | $75.85_{\pm4.37}$ | $75.28_{\pm1.44}$ | $76.31_{\pm1.32}$ | $77.48_{\pm0.97}$ | $86.32_{\pm1.28}$ | $89.30_{\pm1.81}$ | $91.50_{\pm0.48}$ |
| MIWAE | $65.94_{\pm7.68}$ | $69.43_{\pm6.25}$ | $71.70_{\pm3.38}$ | OOM | OOM | OOM | $61.14_{\pm1.81}$ | $64.33_{\pm0.93}$ | $66.11_{\pm0.65}$ |
| GRAPE | $66.88_{\pm3.75}$ | $75.00_{\pm0.81}$ | $54.72_{\pm15.42}$ | $75.50_{\pm1.98}$ | $75.90_{\pm1.35}$ | $77.55_{\pm1.82}$ | $\mathbf{91.80}_{\pm0.41}$ | $91.61_{\pm0.89}$ | $84.57_{\pm18.75}$ |
| IGRM | $68.13_{\pm7.10}$ | $69.33_{\pm8.21}$ | $72.08_{\pm3.85}$ | $75.30_{\pm2.37}$ | $75.13_{\pm1.58}$ | $77.04_{\pm0.85}$ | $62.30_{\pm4.17}$ | $66.38_{\pm1.85}$ | $72.07_{\pm1.83}$ |
| G-FACM | $\mathbf{81.56}_{\pm4.57}$ | $\mathbf{89.00}_{\pm2.71}$ | $\mathbf{84.91}_{\pm4.13}$ | $\mathbf{76.30}_{\pm1.00}$ | $\mathbf{76.62}_{\pm0.67}$ | $\mathbf{77.70}_{\pm3.04}$ | $89.73_{\pm2.32}$ | $\mathbf{91.69}_{\pm1.14}$ | $\mathbf{91.78}_{\pm0.74}$ |

| Dataset | ADNI QT-PAD | | | ADNI TADPOLE | | | Diabetes 130-US | | |
|---|---|---|---|---|---|---|---|---|---|
| Label rate | 5% | 10% | 20% | 5% | 10% | 20% | 5% | 10% | 20% |
| Zero | $78.15_{\pm1.27}$ | $78.22_{\pm0.99}$ | $79.74_{\pm2.21}$ | $72.93_{\pm2.28}$ | $77.94_{\pm1.24}$ | $80.42_{\pm1.44}$ | $50.50_{\pm3.32}$ | $53.66_{\pm0.77}$ | $53.77_{\pm0.98}$ |
| Mean | $78.53_{\pm1.65}$ | $78.35_{\pm1.53}$ | $79.08_{\pm2.61}$ | $71.94_{\pm3.22}$ | $76.89_{\pm1.49}$ | $79.44_{\pm2.00}$ | $52.37_{\pm2.11}$ | $53.76_{\pm0.33}$ | $53.81_{\pm0.67}$ |
| kNN | $79.17_{\pm1.20}$ | $80.39_{\pm1.30}$ | $80.77_{\pm1.53}$ | $72.36_{\pm1.80}$ | $76.68_{\pm1.07}$ | $79.42_{\pm0.97}$ | $51.65_{\pm2.74}$ | $53.92_{\pm0.86}$ | $54.42_{\pm0.28}$ |
| GAIN | $78.76_{\pm1.46}$ | $77.46_{\pm1.22}$ | $79.00_{\pm1.03}$ | $73.07_{\pm0.94}$ | $75.46_{\pm1.22}$ | $80.67_{\pm1.16}$ | $50.60_{\pm3.37}$ | $53.58_{\pm0.59}$ | $53.94_{\pm0.74}$ |
| MIWAE | OOM | OOM | OOM | OOM | OOM | OOM | OOM | OOM | OOM |
| GRAPE | $79.60_{\pm1.85}$ | $79.72_{\pm1.80}$ | $80.11_{\pm1.46}$ | $74.06_{\pm3.09}$ | $77.46_{\pm1.81}$ | $81.04_{\pm1.20}$ | $51.84_{\pm3.57}$ | $53.70_{\pm0.95}$ | $54.02_{\pm0.90}$ |
| IGRM | $78.45_{\pm2.11}$ | $78.60_{\pm1.44}$ | $79.59_{\pm2.52}$ | $73.61_{\pm1.10}$ | $76.80_{\pm1.59}$ | $79.93_{\pm1.38}$ | $50.79_{\pm3.05}$ | $53.49_{\pm0.74}$ | $53.96_{\pm0.67}$ |
| G-FACM | $\mathbf{83.87}_{\pm0.58}$ | $\mathbf{83.75}_{\pm0.99}$ | $\mathbf{85.39}_{\pm0.90}$ | $\mathbf{77.75}_{\pm1.68}$ | $\mathbf{80.01}_{\pm1.22}$ | $\mathbf{81.77}_{\pm0.92}$ | $\mathbf{53.85}_{\pm0.50}$ | $\mathbf{54.65}_{\pm1.11}$ | $\mathbf{54.54}_{\pm1.84}$ |

Table 4: Classification performance measured by Micro-F1 score (%). Standard deviation errors are given.

| Method | Echocardiogram | Duke Breast Cancer | ABIDE | ADNI QT-PAD | ADNI TADPOLE | Diabetes 130-US |
|---|---|---|---|---|---|---|
| G-FACM using GCNMF | $88.33_{\pm2.11}$ | $76.89_{\pm0.94}$ | $81.08_{\pm14.91}$ | $83.31_{\pm1.35}$ | $78.58_{\pm0.67}$ | $53.40_{\pm1.61}$ |
| G-FACM using PaGNN | $88.33_{\pm1.83}$ | $\mathbf{77.17}_{\pm1.69}$ | $90.97_{\pm2.01}$ | $\mathbf{83.85}_{\pm0.61}$ | $80.13_{\pm1.18}$ | $54.41_{\pm1.05}$ |
| G-FACM using PCFI | $87.00_{\pm1.94}$ | $76.29_{\pm1.13}$ | $91.19_{\pm1.31}$ | $83.52_{\pm0.65}$ | $\mathbf{80.68}_{\pm1.48}$ | $52.63_{\pm0.96}$ |
| G-FACM using FP (default) | $\mathbf{89.00}_{\pm2.71}$ | $76.62_{\pm0.67}$ | $\mathbf{91.69}_{\pm1.14}$ | $83.75_{\pm0.99}$ | $80.01_{\pm1.22}$ | $\mathbf{54.65}_{\pm1.11}$ |

graph construction has greatly contributed to adapting graph data imputation methods to medical tabular data, leading to their remarkable performance. Furthermore, it suggests that the superior performance of these methods arises not from simply using a kNN graph, but specifically from utilizing our feature-attentive kNN graph.

### 4.6 Effect of Label Rate on Performance

Since the preliminary training stage, which affects the graph construction stage, utilizes the labels of training samples, the performance of G-FACM may be influenced by the proportion of labeled training samples. Therefore, we compare the average Micro-F1 score of G-FACM and other methods by varying the label rates. Table 3 presents the comparison results for varying label rates. As shown in the table, in all cases except for the ABIDE dataset at a label rate of 5%, G-FACM consistently achieves state-of-the-art performance over tabular imputation methods. These results validate the robustness of G-FACM against varying label rates.

### 4.7 G-FACM using other graph data imputation methods

As mentioned in Sec. 4.2, we utilize FP as the default setting for the graph data imputation method in G-FACM. However, other graph data imputation methods, including GCNMF, PaGNN, and PCFI, can also be employed as the graph data imputation method in G-FACM. To demonstrate that G-FACM using graph data imputation methods other than FP is also effective in medical classification on tabular datasets, we conduct comparative experiments using G-FACM with different graph data imputation methods. Table 4 presents the results of the comparative experiments. As shown in the table, G-FACM models with different graph data imputation methods exhibit competitive classification performance when compared to each other across datasets. The graph data imputation method that achieves the best performance varies depending on the dataset, with each method performing best on a specific dataset. This implies that the outstanding performance of G-FACM does not stem from the use of FP as a graph data imputation method, and other imputation methods can be used in its place. We select FP as the default graph data imputation method in G-FACM because G-FACM using PCFI shows good performance across datasets. However, G-FACM using other graph data imputation methods also generally achieves state-of-the-art performance when compared to the results of existing tabular data imputation methods presented in Table 1. Thus, replacing FP with other graph data imputation methods does not significantly affect the superiority of G-FACM.

## 4.8 Time Complexity Analysis

Here we discuss the time complexity of G-FACM. The time complexity of feature-attentive kNN graph construction consisting of the two stages, the preliminary training stage and the graph construction stage, is $O\left(N \cdot (F^2 + \sum_{l=1}^{L} d_{l-1} \cdot d_l) + N^2 \cdot F\right)$. We then determine the duration of feature-attentive kNN graph construction by measuring running times. Table 5 shows a comparison of the running times among all the methods compared in this paper. We select the Echocardiogram and ABIDE datasets since deep learning-based tabular

Table 5: Running times (seconds). ATT_kNN denotes feature-attentive kNN graph construction.

| Dataset | Echocardiogram | | ABIDE | |
|---|---|---|---|---|
| Method | ATT_kNN | Total | ATT_kNN | Total |
| Zero | - | 4.2 | - | 4.6 |
| Mean | - | 4.3 | - | 4.8 |
| kNN | - | 4.2 | - | 4.6 |
| GAIN | - | 8.5 | - | 13.2 |
| MIWAE | - | 6.4 | - | 20.4 |
| GRAPE | - | 200.1 | - | 721.6 |
| IGRM | - | 1683.3 | - | 1718.4 |
| G-FACM using GCNMF | 1.3 | 7.2 | 1.4 | 10.9 |
| G-FACM using PaGNN | 1.3 | 5.7 | 1.4 | 6.9 |
| G-FACM using PCFI | 1.3 | 11.2 | 1.4 | 12.7 |
| G-FACM using FP (default) | 1.3 | 5.8 | 1.4 | 7.5 |

data imputation methods lead to out-of-memory errors on the other datasets. We observe that feature-attentive kNN graph construction occupies a relatively small portion of the running times in G-FACM. Additionally, we confirm that G-FACM generally take less time compared to deep learning-based tabular imputation methods. In summary, feature-attentive kNN graph construction is a fast algorithm, avoiding any significant time burden on graph data imputation methods. Furthermore, we confirm that G-FACM are more efficient on medical tabular data compared to existing state-of-the-art methods, as shown in Table 1.

## 4.9 Memory Complexity Analysis

In the process of feature-attentive kNN graph construction, the memory is utilized for training the model $g_\theta$ and constructing the $k$NN graph. To mitigate the heavy memory usage during $k$NN graph construction, we leverage a batch-wise kNN graph construction strategy. When constructing kNN graphs among samples, we divide batches with batchsize $B$, and calculate k-nearest neighbors for each batch. This strategy reduces the memory requirement because it avoids the need to store distances between all samples

Table 6: Memory usage of G-FACM for different datasets, measured in gigabytes (GB).

| Dataset | Graph Construction | Total |
|---|---|---|
| Echocardiogram | 0.001 | 1.192 |
| Duke Breast Cancer | 0.506 | 1.338 |
| ABIDE | 0.054 | 1.251 |
| ADNI QT-PAD | 0.734 | 1.572 |
| ADNI TADPOLE | 0.628 | 1.495 |
| Diabetes | 1.326 | 2.213 |

in the entire dataset at once. Specifically, in terms of memory complexity, batch-wise kNN graph construction changes the typical $O(N^2 \cdot F)$ complexity to $O(B \cdot N \cdot F)$. Therefore, the memory complexity of the feature-attentive kNN graph construction process is $O(\theta) + O(B \cdot N \cdot F) + O(N^2)$, where $O(N^2)$ is required for $\mathbf{A}$. We further measure the memory usage in the feature-attentive kNN graph construction process for each dataset. Table 6 shows the results of the measurement. As shown in the table, feature-attentive kNN graph construction requires only a small amount of memory. Furthermore, we can confirm that the entire process of G-FACM, including training GNN models in the final training stage, operates with the reasonable memory usage.

## 4.10 Scalability of G-FACM

To demonstrate G-FACM's scalability, we conduct additional experiments on Diabetes CDC, a large-scale dataset containing 253,680 samples. Since this dataset does not contain missing values initially, we randomly mask 30% of the feature entries to simulate missingness. As shown Table 7, state-of-the-art graph-based imputation methods, including MIWAE, GRAPE, and IGRM, encounter out-of-memory (OOM) issues on this dataset due to their high memory demands. In contrast, G-FACM successfully operates on this large dataset without any memory issues and achieves the best performance among all methods. These results demonstrate that G-FACM is not only effective but

Table 7: Classification performance on Diabetes CDC (%). OOM denotes an out-of-memory error.

| Method | Micro-F1 |
|---|---|
| Zero | $80.84_{\pm 6.44}$ |
| Mean | $84.07_{\pm 0.04}$ |
| kNN | $84.06_{\pm 0.07}$ |
| GAIN | $84.08_{\pm 0.08}$ |
| MIWAE | OOM |
| GRAPE | OOM |
| IGRM | OOM |
| G-FACM | $\mathbf{84.26}_{\pm 0.02}$ |

also scalable to large tabular datasets, highlighting G-FACM's practical applicability in real-world settings involving large-scale medical data.

### 4.11 DOES FEATURE-WISE IMPORTANCE REALLY CAPTURE CLASS-DISCRIMINATIVE FEATURES?

To confirm that the feature-wise importance $\mathbf{t}$ of G-FACM effectively captures class-discriminative features, we conduct an in-depth analysis of $\mathbf{t}$. We extract the two features with the highest values in $\mathbf{t}$ on the ADNI TADPOLE dataset. The features identified are CDRSB_bl and LDELTOTAL_BL, which represent the total score of Clinical Dementia Rating (CDR) and the Logical Memory II Delayed Recall test, respectively, the latter being part of the Wechsler Memory Scale. To verify that these features are class-discriminative features, we calculate the mean and standard deviation of each feature across classes. Each sample in the ADNI TADPOLE dataset belongs to one of five classes related to cognitive impairment: Cognitively Normal (CN), Significant Memory Concern (SMC), Early Mild Cognitive Impairment (EMCI), Late Mild Cognitive Impairment (LMCI), and Alzheimer's Disease (AD). These classes are ordered according to the increasing severity of cognitive impairment, with AD being the most severe.

Table 8: Mean and standard Deviation of the two features with the highest values in $\mathbf{t}$ across different classes. "Std." denotes standard deviation.

| Feature | CDRSB_bl | | LDELTOTAL_BL | |
|---|---|---|---|---|
| Class | Mean | Std. | Mean | Std. |
| CN | 0.003 | 0.012 | 0.578 | 0.146 |
| SMC | 0.005 | 0.015 | 0.565 | 0.145 |
| EMCI | 0.127 | 0.076 | 0.391 | 0.081 |
| LMCI | 0.164 | 0.091 | 0.169 | 0.115 |
| AD | 0.443 | 0.165 | 0.060 | 0.082 |

Table 8 shows the distribution of the two features with the highest values in $\mathbf{t}$. As the severity of cognitive impairment increases, CDRSB_bl increases while LDELTOTAL_BL decreases. This indicates that the values of CDRSB_bl and LDELTOTAL_BL can significantly aid in distinguishing between classes. Furthermore, $\mathbf{t}$ can provide medical insights into which features are critical for disease diagnosis. Conversely, we examine the two features with the smallest values in $\mathbf{t}$. The features found are Years_bl and SITE, which represent the years from the first measurement and an indicator that denotes the specific clinical site where each participant was enrolled, respectively. Table 9 shows the distribution of these two features. We observe that it is difficult to identify trends related to the severity of cognitive impairment in these two features. In summary, the feature-attentive kNN graph constructed in G-FACM effectively captures class-discriminative features with $\mathbf{t}$ and makes the generated kNN graph attentive to class-discriminative features.

Table 9: Mean and standard Deviation of the two features with the lowest values in $\mathbf{t}$ across different classes. "Std." denotes standard deviation.

| Feature | Years_bl | | SITE | |
|---|---|---|---|---|
| Class | Mean | Std. | Mean | Std. |
| CN | 0.122 | 0.215 | 0.101 | 0.186 |
| SMC | 0.072 | 0.137 | 0.067 | 0.055 |
| EMCI | 0.184 | 0.236 | 0.079 | 0.118 |
| LMCI | 0.107 | 0.189 | 0.117 | 0.211 |
| AD | 0.090 | 0.141 | 0.089 | 0.124 |

Appendix B provides additional experiments, including hyperparameter sensitivity (Appendix B.1), the statistical significance of ATT_kNN over kNN (Appendix B.2), feature selection using feature-importance scores (Appendix B.3), analysis of zero initialization (Appendix B.4), and the robustness of ATT_kNN to feature noise (Appendix B.5).

## 5 CONCLUSION

In this paper, we introduce G-FACM, a novel framework for medical tabular data, which seamlessly integrate graph data imputation methods with medical tabular data. While graph data imputation methods have not been considered in the tabular domain, G-FACM provides kNN graphs tailored to these imputation methods. By using G-FACM, these methods transfer their outstanding performance in the graph domain to the medical domain, leading to remarkable performance gains over existing tabular imputation methods. Our work demonstrates the potential for graph data imputation methods to be extended to non-graph-structured data. Furthermore, we believe that our work will contribute to machine learning-based disease diagnosis by significantly improving patient classification performance, thereby supporting more reliable AI-driven healthcare applications.

## Reproducibility Statement

We have made careful efforts to ensure the reproducibility of our work. Specifically, Sec. 4.3 describes the experimental setup, including training/validation/testing splits, the downstream model, and the hyperparameter search range. Detailed descriptions of the datasets, including download sources, are provided in Appendix A.1, while implementation details are presented in Appendix A.2. In addition, Appendix C includes URL links to all baseline implementations. The complete source code will be made publicly available upon publication.

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

# A EXPERIMENTAL DETAILS

Table 10: Dataset Statistics. $N$ and $F$ denote the number of samples and features, respectively. $F_{num}$ and $F_{cat}$ represent the number of numerical features and categorical features, respectively. We transform numerical features by scaling them to a fixed range between 0 and 1. We utilize one-hot encoding for categorical features. While $C$ represents the number of classes, $r_m$ denotes the missing rate of features in a given dataset.

| Dataset | $N$ | $F$ | $F_{num}$ | $F_{cat}$ | $C$ | $r_m$ |
|---|---|---|---|---|---|---|
| Echocardiogram | 74 | 12 | 3 | 9 | 2 | 2.59% |
| Duke Breast Cancer | 907 | 93 | 34 | 59 | 2 | 11.94% |
| ABIDE | 1112 | 104 | 85 | 19 | 2 | 52.52% |
| ADNI QT-PAD | 1737 | 96 | 76 | 20 | 5 | 22.29% |
| ADNI TADPOLE | 2132 | 110 | 89 | 21 | 5 | 27.31% |
| Diabetes 130-US | 10177 | 47 | 11 | 36 | 3 | 4.03% |
| Diabetes CDC | 253680 | 21 | 21 | 0 | 3 | 30.00% |

Table 11: Hyperparameter settings of G-FACM for each graph data imputation method across different datasets.

| Method | Hyperparameter | Echocardiogram | Duke Breast Cancer | ABIDE | ADNI QT-PAD | ADNI TADPOLE | Diabetes 130-US | Diabetes CDC |
|---|---|---|---|---|---|---|---|---|
| GCNMF | $k$ | 10 | 5 | 10 | 10 | 5 | 5 | - |
| | $\alpha$ | 1 | 0.25 | 1 | 0.75 | 0.5 | 1 | - |
| PaGNN | $k$ | 10 | 10 | 1 | 1 | 10 | 1 | - |
| | $\alpha$ | 1 | 1 | 0.5 | 0.5 | 0.25 | 1 | - |
| PCFI | $k$ | 10 | 10 | 1 | 10 | 10 | 1 | - |
| | $\alpha$ | 1 | 0.5 | 0.5 | 0.75 | 0.75 | 1 | - |
| FP (default) | $k$ | 10 | 3 | 1 | 1 | 5 | 5 | 10 |
| | $\alpha$ | 0.5 | 0.25 | 0.5 | 0.5 | 0.25 | 1 | 0.5 |

## A.1 DATASET DETAILS

We conduct experiments on seven benchmark datasets, including Echocardiogram, Duke Breast Cancer, ABIDE, ADNI QT-PAD, ADNI TADPOLE, Diabetes 130-US, and Diabetes CDC. Table 10 presents the statistics of the datasets used in this paper.

### A.1.1 ECHOCARDIOGRAM

The Echocardiogram dataset is a medical tabular dataset related to heart attacks, which can be downloaded from the UCI Machine Learning Repository (Asuncion et al., 2007). The 'alive-at-1' feature, a binary variable, is used as the class label. In this label, 0 indicates that the patient either died within one year or was followed for less than one year, while 1 indicates that the patient was alive at one year. The goal of the classification problem using the Echocardiogram dataset is to predict whether patients will survive for at least one year after a heart attack.

### A.1.2 DUKE BREAST CANCER

The Duke Breast dataset is a medical tabular dataset related to breast cancer, available for download from The Cancer Imaging Archive (TCIA) (Saha et al., 2018). The 'Tumor_Grade' feature, which can be one of {1, 2, 3}, is used as the class label.

### A.1.3 ABIDE

The Autism Brain Imaging Data Exchange (ABIDE) dataset is a medical tabular dataset related to autism spectrum disorder, available for download from the ABIDE webpage (Di Martino et al., 2014). The 'DX_GROUP' feature, where 1 and 2 represent autism and control, respectively, is used as the class label.

### A.1.4 ADNI QT-PAD AND ADNI TADPOLE

The Alzheimer's Disease Neuroimaging Initiative (ADNI) dataset is a medical tabular dataset used to study the progression of Alzheimer's disease (AD), which can be downloaded from the ADNI webpage (Petersen et al., 2010). We use the 'DB_bl' feature as the class label, which can be one of five

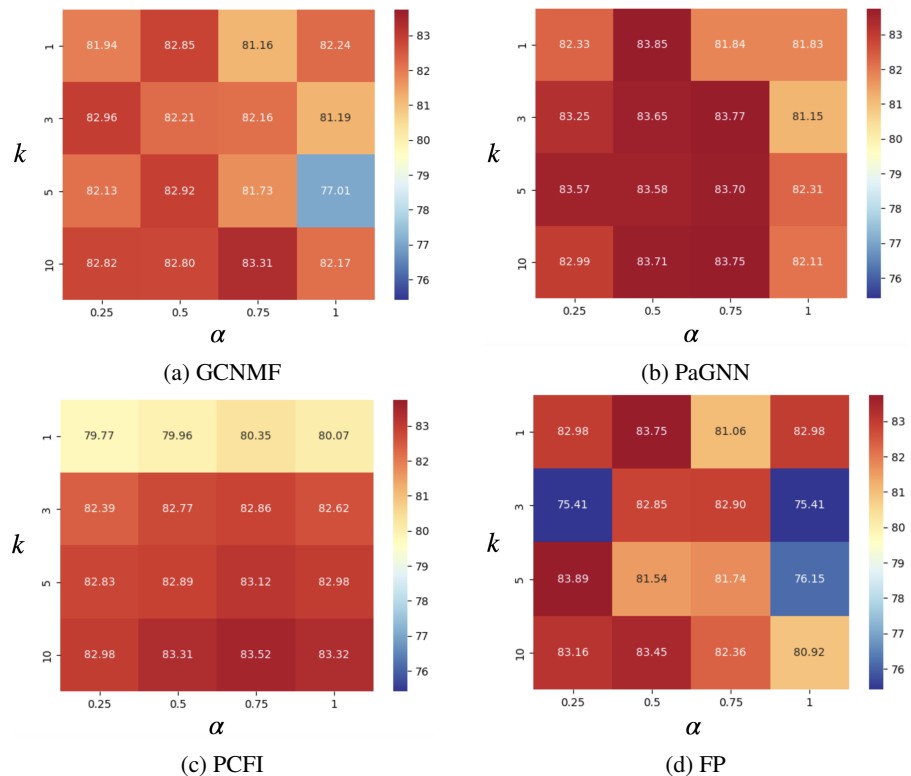

Figure 2: Classification performance of G-FACM for different $k$ and $\alpha$ on the ADNI QT-PAD dataset, measured by Micro-F1 score (%).

cognitive impairment levels: Cognitively Normal (CN), Significant Memory Concern (SMC), Early Mild Cognitive Impairment (EMCI), Late Mild Cognitive Impairment (LMCI), and Alzheimer's Disease (AD). These classes are ordered by increasing severity of cognitive impairment, with AD being the most severe. 'ADNI QT-PAD' and 'ADNI TADPOLE' are located in the tadpole_challenge folder and ADNI_QT-PAD, respectively.

### A.1.5 DIABETES 130-US

The Diabetes 130-US dataset represents the records of hospitalized patients diagnosed with diabetes, which can be downloaded from the UCI Machine Learning Repository (Asuncion et al., 2007). The readmitted feature is the class label, which can be one of the following: 1) '<30': if the patient was readmitted in less than 30 days; 2) '>30': if the patient was readmitted in more than 30 days; 3) 'No': if there was no record of readmission. The goal is to determine whether the patient will be readmitted within 30 days of discharge.

### A.1.6 DIABETES CDC

The Diabetes CDC dataset contains healthcare statistics and lifestyle survey information about the general population, along with diabetes diagnoses, which can be downloaded from the UCI Machine Learning Repository (Asuncion et al., 2007). The target variable for classification indicates whether a patient has diabetes, is pre-diabetic, or is healthy, corresponding to class labels 0, 1, and 2, respectively.

Table 12: $p$-values comparing our ATT_kNN to kNN on each dataset.

| | Echocardiogram | Duke Breast Cancer | ABIDE | ADNI QT-PAD | ADNI TADPOLE | Diabetes 130-US |
|---|---|---|---|---|---|---|
| kNN | $83.26_{\pm 4.66}$ | $74.90_{\pm 2.58}$ | $90.17_{\pm 1.56}$ | $82.81_{\pm 1.18}$ | $79.15_{\pm 0.98}$ | $52.98_{\pm 1.15}$ |
| ATT_kNN | $86.49_{\pm 4.11}$ | $\mathbf{76.28_{\pm 1.18}}$ | $91.31_{\pm 0.91}$ | $\mathbf{83.26_{\pm 1.01}}$ | $80.65_{\pm 1.10}$ | $54.38_{\pm 1.33}$ |
| Improvement | **+3.88%** | **+1.84%** | **+1.26%** | **+0.54%** | **+1.90%** | **+2.64%** |
| $p$-value | $1.77 \times 10^{-5}$ | $3.46 \times 10^{-4}$ | $7.09 \times 10^{-7}$ | $3.60 \times 10^{-6}$ | $1.60 \times 10^{-10}$ | $9.33 \times 10^{-6}$ |

Table 13: Classification performance with different proportions of features selected by G-FACM's feature importance on ADNI TADPOLE, measured by Micro-F1 score (%).

| Feature Selection (%) | 0.1% | 1% | 5% | 10% | 25% | 50% | 75% | no feature selection |
|---|---|---|---|---|---|---|---|---|
| # features | 2 | 29 | 146 | 293 | 733 | 1466 | 2199 | 2932 |
| Micro-F1 | $65.33_{\pm 5.67}$ | $80.82_{\pm 1.76}$ | $80.98_{\pm 0.73}$ | $80.42_{\pm 1.04}$ | $80.57_{\pm 0.89}$ | $80.50_{\pm 0.98}$ | $80.48_{\pm 1.18}$ | $80.01_{\pm 1.22}$ |

### A.2 Implementation Details

We conduct all experiments on a single NVIDIA GeForce RTX 2080 Ti GPU with 11GB of memory and an Intel Core i5-10500 CPU at 3.10GHz. Across all baselines, we adhere to the hyperparameter tuning strategies and settings described in their respective papers. For training graph data imputation methods used in G-FACM, we follow (Rossi et al., 2022). We utilize the Adam optimizer (Kingma & Ba, 2014) and set the maximum number of epochs to 10,000. We employ an early stopping strategy based on validation accuracy, with a patience of 200 epochs. Dropout (Srivastava et al., 2014) is applied with a drop probability $p$, where $p$ is searched within $\{0, 0.5\}$. We consistently set the number of GNN layers and the hidden dimension of graph data imputation methods to 2 and 64, respectively. Table 11 shows the hyperparamter settings of G-FACM for graph data imputation methods across different datasets. For MLP models, the number of layers is set to 2, with hidden dimensions searched within $\{16, 64, 256\}$, respectively. Learning rates are selected within $\{0.0005, 0.005, 0.05\}$ based on validation sets.

## B Additional Experiments

### B.1 Hyperparameter Sensitivity

To investigate the impact of the two hyperparameters of G-FACM, $\alpha$ and $k$, we measure the classification performance of G-FACM using graph data imputation methods by varying $\alpha$ and $k$ on the ADNI QT-PAD dataeset. According to our search ranges for $k$ and $\alpha$, we vary $k$ and $\alpha$ within $\{1, 3, 5, 10\}$ and $\{0.25, 0.5, 0.75, 1.0\}$, respectively. Figure 2a, Figure 2b, Figure 2c, and Figure 2d demonstrate the classification performance of G-FACM using graph data imputation methods for different $k$ and $\alpha$, measured by Micro-F1 score (%). As shown in the figures, the methods generally demonstrate robustness against variations in $k$ and $\alpha$. Considering the previous state-of-the-art performance of kNN is 80.39%, G-FACM using graph data imputation methods achieve the state-of-the-art performance with most combinations of $(k, \alpha)$ within the respective search ranges. For example, G-FACM using PaGNN consistently outperforms the previous state-of-the-art performance, regardless the values of $k$ and $\alpha$.

### B.2 Statistical Significance of ATT_kNN over kNN

As shown in Table 2, when used as a graph construction method in G-FACM, ATT_kNN achieves notable performance gains in Micro-F1 score over simple kNN. To evaluate the statistical significance of these improvements, we conduct paired t-tests across 50 random splits for each dataset. The results are summarized in Table 12, and all resulting $p$-values are far smaller than the commonly used threshold of $p < 0.01$, confirming that the gains are statistically significant across all datasets.

### B.3 Can Feature Importance Be Used For Feature Selection?

We investigate how the feature-wise importance scores produced by G-FACM can be utilized to support clinical decision-making through feature selection. To this end, we conduct additional experiments in which only the top of features with the highest values are retained, and classification

Table 14: Performance comparison of G-FACM with different initialization strategies, measured by Micro-F1 score (%).

| Dataset | ABIDE | | ADNI TADPOLE | |
|---------|-------|---|--------------|---|
| Initialization | Zero (used) | Mean | Zero (used) | Mean |
| G-FACM using GCNMF | $81.08_{\pm14.91}$ | $78.47_{\pm14.84}$ | $\mathbf{78.58}_{\pm0.67}$ | $71.99_{\pm9.05}$ |
| G-FACM using PaGNN | $90.97_{\pm2.01}$ | $75.21_{\pm10.95}$ | $\mathbf{80.13}_{\pm1.18}$ | $80.01_{\pm1.50}$ |
| G-FACM using PCFI | $91.19_{\pm1.31}$ | $81.96_{\pm3.96}$ | $\mathbf{80.68}_{\pm1.48}$ | $80.13_{\pm1.59}$ |
| G-FACM using FP | $91.69_{\pm1.14}$ | $75.80_{\pm11.05}$ | $\mathbf{80.01}_{\pm1.22}$ | $79.55_{\pm1.89}$ |

Table 15: Performance comparison of graph construction methods in G-FACM under different feature noise levels, measured by Micro-F1 score (%).

| Dataset | ABIDE | | | | ADNI TADPOLE | | | |
|---------|-------|--------|-----|---------|--------------|--------|-----|---------|
| $\sigma$ | FC | SIM_FC | kNN | ATT_kNN | FC | SIM_FC | kNN | ATT_kNN |
| 0 | $49.62_{\pm1.80}$ | $49.62_{\pm1.80}$ | $90.65_{\pm1.51}$ | $\mathbf{91.69}_{\pm1.14}$ | $28.96_{\pm0.57}$ | $28.96_{\pm0.57}$ | $78.90_{\pm0.94}$ | $\mathbf{80.01}_{\pm1.22}$ |
| $10^{-1}$ | $49.62_{\pm1.80}$ | $49.62_{\pm1.80}$ | $90.65_{\pm1.51}$ | $\mathbf{91.69}_{\pm1.14}$ | $28.96_{\pm0.57}$ | $28.96_{\pm0.57}$ | $78.90_{\pm0.94}$ | $\mathbf{79.25}_{\pm1.10}$ |
| $10^{-0.5}$ | $49.62_{\pm1.80}$ | $49.62_{\pm1.80}$ | $88.65_{\pm2.30}$ | $\mathbf{89.80}_{\pm1.38}$ | $28.96_{\pm0.57}$ | $28.96_{\pm0.57}$ | $70.67_{\pm2.16}$ | $\mathbf{72.66}_{\pm0.90}$ |
| 1 | $49.62_{\pm1.80}$ | $49.62_{\pm1.80}$ | $75.33_{\pm2.29}$ | $\mathbf{80.97}_{\pm4.21}$ | $28.96_{\pm0.57}$ | $28.96_{\pm0.57}$ | $32.06_{\pm4.64}$ | $\mathbf{37.17}_{\pm1.53}$ |

performance is evaluated using these selected features. Table 13 presents the results of this feature selection experiment on ADNI TADPOLE. As shown in the table, using only the top 1% of features (29 out of 2932) already achieves a Micro-F1 score of $80.82 \pm 1.76$, which outperforms the full-feature case ($80.01 \pm 1.22$). This indicates that G-FACM effectively identifies a small subset of highly informative features, and that these importance scores can be leveraged for compact and interpretable feature selection. This capability holds strong potential for clinical decision support, where focusing on a few key biomarkers is often critical. Although our primary focus is not on feature selection, these results clearly demonstrate the potential of G-FACM to be utilized for this purpose in clinical decision support.

### B.4 WHY IS ZERO INITIALIZATION USED FOR MISSING VALUES?

In the graph construction stage of G-FACM, we use $\overline{\mathbf{X}}$, obtained from $\mathbf{X}$ by imputing missing values with zeros, *i.e.*, we use zero initialization. To justify this initialization strategy, we conduct comparative experiments using a different initialization strategy. We select mean imputation as the comparison strategy, which is a commonly used strategy for initializing missing values. Mean imputation fills in missing values with the mean of observed values. Table 14 shows the results on the ABIDE and ADNI TADPOLE datasets. As shown in the table, G-FACM using zero initialization consistently outperforms that using mean initialization across the graph data imputation methods on both datasets. These performance gains with zero initialization are attributed to the inherent characteristics of medical tabular data, which often contains many zero values. For instance, among the observed values in the ABIDE and ADNI TADPOLE datasets, 68.10% and 94.78%, respectively, are zeros. This prevalence of zero values makes zero initialization effective in the graph construction stage of G-FACM on medical tabular datasets.

### B.5 ROBUSTNESS OF ATT_kNN TO FEATURE NOISE

To evaluate the robustness of the feature-attentive kNN graph against feature-level noise, we conduct additional experiments where Gaussian noise with varying standard deviations ($\sigma \in \{0, 10^{-1}, 10^{-0.5}, 1\}$) is injected into input features. We compare the performance of G-FACM under different construction methods (fully-connected (FC), similarity-weighted FC (SIM_FC), vanilla kNN, and ATT_kNN). As shown Table 15, G-FACM using ATT-kNN consistently outperforms the other graph construction methods across all noise levels, showing significantly smaller degradation in classification performance as $\sigma$ increases. This confirms that the feature-attentive kNN graph is more robust to input noise compared to the other methods.

# C  Licenses and Repositories of Baseline Methods

Table 16: URL links and license information for the baseline methods used in our experiments.

| Baseline | URL link | License |
|---|---|---|
| GAIN | https://github.com/vanderschaarlab/hyperimpute/blob/main/src/hyperimpute/plugins/imputers/plugin_gain.py | MIT |
| MIWAE | https://github.com/vanderschaarlab/hyperimpute/blob/main/src/hyperimpute/plugins/imputers/plugin_miwae.py | MIT |
| GRAPE | https://github.com/maxiaoba/GRAPE | MIT |
| IGRM | https://github.com/G-AILab/IGRM | Not specified |
| GCNMF | https://github.com/marblet/GCNmf | MIT |
| PaGNN | https://github.com/twitter-research/feature-propagation | Apache-2.0 |
| FP | https://github.com/twitter-research/feature-propagation | Apache-2.0 |
| PCFI | https://github.com/daehoum1/pcfi | Apache-2.0 |

