# OpenReview forum: "Harnessing Patient Connectivity for Medical Classification under Missing Values"
_ICLR.cc/2026/Conference — ICLR 2026 Conference Withdrawn Submission_

### Official Review · Reviewer_U2De · 2025-10-27

**Soundness:** 3
**Presentation:** 3
**Contribution:** 2
**Rating:** 4
**Confidence:** 3

**Summary:**

This paper proposes a method for patient classification in medical table data with missing values. Each patient is treated as a node, and connections are established between patients based on feature similarity, enabling the use of a GNN to process the original table data. This method was validated on six real-world medical table datasets (e.g., Echocardiogram, Duke Breast Cancer, and ADNI), achieving promising results.

**Strengths:**

1. The article has some innovations. It regards each patient as a node and establishes connections between patients through feature similarity, so that GNN can be used to process the original tabular data.
2. The experiments in the article are sufficient in number, and the structure diagram is clear.

**Weaknesses:**

1. The task of the article is a classification task. The author only used the Micro-F1 score as the main performance indicator, without reporting other indicators such as Accuracy, Precision, Recall, AUC, etc.
2. In memory complexity analysis, comparisons should be made with other models, such as IGRM[2] and GRAPE[1].
3. The article uses an imputation method. MAE should be used to measure the difference between the predicted and completed values.
4. G-FACM requires multiple steps: training an MLP classifier with feature attention to obtain feature importance weights, then constructing the kNN graph, and finally training. The article's method seems overly complex compared to end-to-end training, such as IGRM[2].


[1].You, J., Ma, X., Ding, Y., Kochenderfer, M. J., & Leskovec, J. (2020). Handling missing data with graph representation learning. Advances in Neural Information Processing Systems, 33, 19075-19087.
[2]. Zhong, J., Gui, N., & Ye, W. (2023, June). Data imputation with iterative graph reconstruction. In Proceedings of the AAAI Conference on Artificial Intelligence (Vol. 37, No. 9, pp. 11399-11407).

**Questions:**

1. The experiments are conducted with a relatively low label rate (5–20%). However, related works like IGRM report results under higher label rates (e.g., 70%). Including experiments at similar levels would enable a more balanced and comprehensive comparison.
2. Is the G-FACM graph (feature-attentive kNN graph) static? Does it remain fixed throughout the training process?
3. Why is only the Micro-F1 score used for classification tasks, without Accuracy, Precision, Recall, AUC, etc?

---

### Official Review · Reviewer_pBPr · 2025-10-31

**Soundness:** 2
**Presentation:** 3
**Contribution:** 1
**Rating:** 2
**Confidence:** 4

**Summary:**

This manuscript proposes a framework called G-FACM, aiming to apply graph data imputation methods to the (medical) tabular data classification task. The core innovation lies in constructing a feature attention kNN graph, which is weighted by feature importance obtained from a pre-trained MLP, thereby connecting patients with similar class-discriminative features. Although experimental results achieve sota on these datasets, I have negative opinions regarding the paper's methodological novelty, the rationality of its core assumptions, and the fairness of the experimental comparisons.

**Strengths:**

1. The paper addresses the significant and practical problem of classification on medical tabular data with missing values, which is a highly relevant task.

2. The proposed method demonstrates strong empirical performance, achieving state-of-the-art (SOTA) classification results across multiple benchmark datasets.

**Weaknesses:**

1. The core contribution, "first attempt to apply graph data imputation methods to tabular data", is exaggerated and factually incorrect. The cited works, IGRM and GRAPE, are already taking tabular dataset into a bipartite graph to handle missing data.

2. The performance improvement of the proposed method in the manuscript is questionable. The authors used an MLP classifier in the baseline, while G-FACM used a GNN as the classifier. The performance improvement may simply be because the GNN classifier is inherently stronger than the MLP classifier (after the data is constructed into a graph).

3. Based on the high prevalence of '0' value in some medical data, the authors argue that using '0' to fill missing values ​​is more reasonable than using the 'mean'. I find this reasoning insufficient. The Echocardiogram dataset has a missing value rate of only 2.59%, yet G-FACM achieves a classification accuracy of 89%. Does this suggest that the imputation method is fundamentally unimportant in the design of G-FACM?

**Questions:**

see weakness.

---

### Official Review · Reviewer_8cXP · 2025-11-01

**Soundness:** 2
**Presentation:** 2
**Contribution:** 1
**Rating:** 2
**Confidence:** 3

**Summary:**

The authors tackle the challenge of missing data imputation within the medical field. To this end, the authors select six medical datasets with varying amounts of missingness. The authors propose using graph missingness imputation methods to address the challenge of imputing missing values in tabular data. The authors propose to use the F1 classification score on unseen data as a proxy for evaluating the performance of the imputation methods.

The authors claim the following contributions:
1. First attempt to apply graph data imputation methods to tabular data.
2. Novel G-FACM builds a kNN graph that is attentive to class-discriminative features, bridging graph data imputation methods and medical tabular data.
3. G-FACM using feature-attentive kNN graphs significantly outperforms existing state-of-the-art methods in medical tabular classification,
4. G-FACM can also provide valuable medical insights.

**Strengths:**

# Originality
The proposes method is original as claimed by the authors

# Quality
The presentation is good and of high quality

# Clarity
The paper is well written and easy to follow

#Significance

The results are significant to the medical domain, but the general applicability of the method is not apparent.

**Weaknesses:**

# Experiments are insufficient

The exist several libraries that tackle missingness imputation. These libraries evaluate the impact along several methods, ranging from supervised to unsupervised utility, as well as various quality metrics. It would be beneficial to see experiments that detail these metrics.

Of the novel contributions, only those relevant to the ICLR community are considered. Those are the contributions that are domain agnostic.
It would have been valuable to see the approach on synthetic data with induced missingness, as well as in other domains, to truly evaluate the effectiveness of the method.

**Questions:**

How does this method perform compared to the others as the amount of misclassification is varied between extremes?
How does it perform in terms of actual missingness implication?
Can we assess how well the missing values and the imputed values align with each other?

---

### Official Review · Reviewer_ZrZ6 · 2025-11-01

**Soundness:** 1
**Presentation:** 3
**Contribution:** 2
**Rating:** 2
**Confidence:** 3

**Summary:**

The paper presents a method for tabular data imputation using a two-step process. First, a network is built from the samples, where samples with a high probability of being in the same class share a stronger relation. Second, a graph imputation method is applied to this newly constructed graph. The paper validates this approach on several comparative benchmarks.

**Strengths:**

The paper validates its approach on comparative benchmarks, providing empirical results for the proposed method.

**Weaknesses:**

- The primary weakness is the lack of a clear motivation for the proposed pipeline. It is difficult to follow the claim of how this two-step process (building a graph, then imputing on it) is fundamentally different from a standard imputation method that learns from the data (e.g., one trained on the entire dataset, on a specific class, or on features selected for class discrimination). While learning a graph before applying graph-based methods is a known technique, it usually requires specific motivation, such as leveraging inductive bias like in biological networks. That motivation is absent here. The contribution does not appear to be a sufficiently novel step forward for ICLR.

- The paper's claim in the abstract and title that current methods use only the specific sample to impute missing data seems incorrect. Even simple methods like mean imputation are based on other samples, and more advanced ML-based imputers learn from the entire dataset to model the data's conditional probabilities.

- The paper misses a fundamental difference between its setup and the existing graph imputation literature. In previous work, the graph typically represents additional information or a known inductive bias. Here, the graph is constructed from the data itself (essentially clustering samples from the same class) and then used for imputation. This appears to be a circular approach, and it's not clear how this provides more information than simply using an imputer trained on a single class or on discriminative features.

- The paper does not provide a clear justification for its sophisticated feature selection method (lines 052-053). It is not explained why a standard ML-based method for feature importance would be insufficient for this task.

**Questions:**

"However, most existing techniques classify patients solely based on each patient’s individual features, overlooking the potential benefits of using similarities among patients to improve both imputation and classification."   Can the authors explain why any classifier or regressor would fail to catch the similarity between samples?

---

### Official Review · Reviewer_JkW1 · 2025-11-01

**Soundness:** 3
**Presentation:** 3
**Contribution:** 1
**Rating:** 2
**Confidence:** 4

**Summary:**

This paper proposes a model called Graph-based Feature-Attentive Classifier under Missingness (G-FACM) for medical tabular classification with missing values. It trains a feature-wise attention MLP to estimate per-feature importance, uses those importances to construct a feature-attentive k-NN graph (weighted cosine similarity with hyperparameter $\alpha$), and then feeds the resulting adjacency matrix to graph imputation methods (e.g., FP) to perform node classification. Empirical evaluations are done using six benchmark datasets.

**Strengths:**

- The problem to be adressed is significant. Handling missing data is one of the central problems in medical tabular learning.
- The idea of learning task-specific feature weights and constructing patient-similarity graphs for tabular data is a reasonable design to combine graph imputation/classification and medical tabular data learning.
- The paper uses multiple benchmark datasets to showcase the empirical performance of the proposed method.
- The paper is overall structured and easy to follow.

**Weaknesses:**

- My major concern is that the novelty of the paper is limited. It is argued to be the first attempt to apply graph data imputation methods to tabular data. However, constructing similarity graphs by nearest neighbors and learned distance metrics have been widely applied to provide auxiliary information for tabular learning. One can refer to this survey paper for a summary of such prior works: https://dl.acm.org/doi/full/10.1145/3744918

**Questions:**

1. How does the proposed method differ from existing metric learning approaches or the feature-weighted distance methods like the TabNet feature masks?
2. Are different missing mechanisms considered when constructing the similarity graph?
3. How does the learned attention MLP generalize to unseen patients at inference?

---

### Note · Authors · 2025-11-16

I have read and agree with the venue's withdrawal policy on behalf of myself and my co-authors.